# Assessment Methodology for Physical Vulnerability of Vernacular Architecture in Areas Affected by Depopulation: The Case of Comunidad Valenciana, Spain

Eva Tortajada Montalvá , Camilla Mileto * and Fernando Vegas López-Manzanares

Centro de Investigación en Arquitectura, Patrimonio y Gestión para el Desarrollo Sostenible (PEGASO), Universitat Politècnica de València, 46022 Valencia, Spain; evtormon@upv.es (E.T.M.); fvegas@cpa.upv.es (F.V.L.-M.)
* Correspondence: cami2@cpa.upv.es

**Abstract:** The intensity with which the phenomenon of depopulation has affected rural municipalities in Spain between 1950 and 2022 has led to a loss in the intergenerational transmission of traditional knowledge, values and customs. Sociocultural loss entails associated physical risks: the abandonment, demolition, and loss of vernacular architecture. This research analyzes the evolution of this type of architecture in a period of acute depopulation and its current state of conservation. A total of 180 case studies in the region of Comunidad Valenciana are analyzed through four factors affecting the physical vulnerability of dwellings: year of construction, state of conservation, type of use, and a combination of all three. Data management software is used to combine all the information and produce the results in a tabular and graphical format, while the Geographic Information System is used to draw up risk maps showing the results. These results are then divided into analysis groups created according to the degree of depopulation observed in the years mentioned. This made it possible to identify the relationship between depopulation and the conservation of vernacular architecture, showing the risk level for each case study, and thus creating an analysis methodology applicable in other territories affected by depopulation at a national and international level.

**Keywords:** depopulation; vernacular architecture; risks; vulnerability; dwelling; resilience; sustainable development; assessment methodology

## 1. Introduction

At the beginning of the 21st century, and for the first time ever, the urban population is exceeding the rural population [1]. What some see as a milestone in the history of urbanism, progress and modernity, others view as a reflection of the great cultural loss caused by emigration from the countryside to the city [2]. The year 2020 represented a milestone in the world's demographic history, as the global rural population decreased for the first time. While the current trend is for higher numbers of people living in urban areas, rural areas are also becoming less and less populated. In Europe, since 1950, the population decrease observed has been less than 1% each year. Since 2020, however, the decline in rural areas in Europe has exceeded this 1% threshold. According to European statistics, some countries with historically high rurality rates are currently suffering significant depopulation. This is the case, for instance, of Poland (−2.6% annual rural population growth), Bulgaria (−7.6% annual rural population growth) and the Netherlands (−3.4% annual rural population growth) [3].

Spanish rural territories, known as Spanish Lapland [4], form the largest demographic desert in the European Union as a result of depopulation. Between 1900 and 2001, the demographic evolution of the rural population in Spain in relation to the total population followed a negative trend. In 1900, 68% of Spanish residents lived in municipalities of under 10,000 inhabitants. One hundred years later, in 2001, less than 25% of the population lived in

rural areas. This variation in the distribution of population is accompanied by a variation in the number of rural municipalities. In 1900, there were approximately 7900 municipalities of under 10,000 inhabitants, as opposed to the less than 7500 in 2001 [5]. At present, in Spain, there are 7367 municipalities of less than 10,000 inhabitants (out of a total of 8131), accounting for 90% of the total municipalities in the country.

The situation is much more serious in the smaller rural municipalities, considered as those with a population of less than 5000 inhabitants [6]. According to the 2021 census, the population in these municipalities was 5,687,092 in total, which is equivalent to 12% of the Spanish population. Paradoxically, this population is distributed throughout 69.3% of the territory [7].

Depopulation is a demographic and territorial phenomenon that consists of a decrease in the number of inhabitants of a given territory or nucleus compared to an earlier period. The decrease in absolute terms of the number of inhabitants can be the result of a negative population growth (when deaths exceed births), a negative migratory balance (emigration exceeds immigration), or both simultaneously. The two phases are identified in the depopulation phenomenon in Spain: between 1950 and 1991, depopulation was mainly due to negative migratory balance, whereas from 1991 onwards, depopulation was primarily the result of negative population growth [8]. This situation will not change in the near future, as exponential growth throughout the country is needed in order to recover.

The phenomenon of depopulation must be addressed through an interpretation of rural development that is geared toward a more qualitative holistic prosperity, with comprehensive management and an interdisciplinary analysis [9]. Depopulation and development must be jointly articulated based on human, social and relational capital, eschewing strategies based solely on material, human and economic growth. In order to guarantee effective management, culture is a vital cross-disciplinary element granting a social dimension to these policies [10].

Depopulation has had a profound impact on rural society, leading to the end of traditional life for the vast majority of its inhabitants. This phenomenon has brought about a radical transformation in the dynamics of rural communities, with significant repercussions in their social, economic and cultural structure [5]. Culture encompasses all the products generated by humans, covering both tangible and intangible aspects, including society as an intangible creation. Society is destined to structure relations between individuals, and it is through this interaction between humans that it is formed and maintained [11]. In order to maintain and renew a culture, society must transmit the cultural meanings and beliefs from one generation to the next. Whenever a social or intellectual change such as depopulation occurs, new needs arise that threaten the continuity of the group [12]. This can cause the transmission of traditional culture to be interrupted, so that society begins to lose its subjective reality.

A resilient society can adapt to medium–long-term changes, as it can forecast, respond, recover and learn. However, the shock [13] caused by the phenomenon of depopulation, occurring so rapidly in such a short space of time, has not given society a chance to adapt and recover. The phenomenon of depopulation in Spain, so intense between 1950 and 1991, brought about the incapacity of rural societies to transmit the customs, ways of life and knowledge acquired over time. The rupture between generations, the excessively quick change, and the disorder resulting from depopulation in these societies risk the disappearance of the entire traditional cultural heritage. The disappearance of cultural traditions, natural and cultural settings, identities and values, results in the loss of all the knowledge acquired over centuries, and, if these are not known, it will never be possible to fully comprehend a society [14,15].

The cultural loss observed in rural societies is irreversible in most cases. All cultural expressions have taken place in a given space, so the delimitation and organization of this space is a necessary prerequisite for any type of cultural representation. Therefore, built surroundings are one of the main elements in the culture of a place. It is the physical representation of the nature and organization of a society, its culture, values and symbols,

the social and spatial distribution of labor, and the political and economic formation on which this society was based [16,17]. In this context, vernacular architecture that has survived the phenomenon of depopulation becomes a highly valuable cultural element to be preserved, as it expressively embodies the values of traditional societies [18,19].

Spanish rural population accounts for 90% of the territory. This is home to a significant part of the country's cultural heritage [6], and almost the entire vernacular heritage in the country. This vast environmental, natural, social and cultural heritage, part of which has not yet been lost, is still largely preserved in rural and mountainous areas and could enable these territories to overcome the challenges facing them in the near future [20]. These territories, which are the ones most affected by depopulation, also risk their own construction culture falling into oblivion as a result of not having been transmitted to successive generations [21].

While the values of vernacular architecture are recognized, many risks have been identified that could lead to its disappearance. In terms of depopulation and rural dwellings, many factors, including the insalubrious conditions of rural dwellings, lead to an increased risk of emigration from the countryside to the city [22], and in turn to the risk of the abandonment of vernacular architecture [23,24]. This abandonment by inhabitants hampers its conservation [25–29], and the risk increases with the lack of attached value, the loss of traditions and changes in use on the part of new generations [27,29–33] (p. 69, [34]). This lack of intergenerational transmission has resulted in a loss of traditional knowledge and crafts [35–37], increased by a lack of knowledge and sensitivity on the part of owners, architects and authorities [27,37,38] (pp. 60–63, [34]). These risks are mostly linked to new industrial practices, which entail the loss and oblivion of local work and traditional and artisanal crafts [23,27,36,38–43]. At the same time, speculation in architecture [27,30,44–46] and the rise in tourism [26,30,47] often lead to the demolition, imitation, vulgarization and homogenization of vernacular architecture [26,27,30,32,44,45,47].

The phenomenon of depopulation has different consequences, the most predominant of which are of a negative nature. On the one hand, the lack of intergenerational transmission has resulted in significant cultural loss, particularly as regards intangible culture. On the other hand, it should be noted that there is a positive aspect in relation to tangible culture. Although many constructions have been lost completely due to decay or poor practice, their locations being far from centers of power, the abandonment of some of these traditional constructions has ensured the conservation of a considerable number of vernacular constructions in Spanish villages. Thus, it has become possible to recover, conserve and transmit the values of traditional societies through their constructive legacy [24,48–55].

Emphasis is placed on the role of rural architectural heritage and its potential as a resource for sustainable development in these regions, as well as on the need to preserve, recover and conserve traditional rural architecture. A suitable intervention can reactivate traditional crafts, boost the local economy, fight against depopulation, strengthen cultural identity, protect the territory and natural landscape, and promote sustainable development. Thus, by carrying out actions that respect the vernacular setting, the place can be revitalized [52]. In order to recover and conserve the vernacular architecture of a place, it is first necessary to ascertain its current condition. Thereby, solutions can be adapted to the risks and threats of each place.

Most of the studies on the current condition of vernacular architecture focus on a small scale (village, municipality or region) and tend to analyze specific aspects (material condition, state of conservation and construction techniques). Several methodologies have been developed for the analysis of vulnerability factors of vernacular architecture. These include seismic vulnerability [56–58], earthen vernacular architecture vulnerability [59,60] and multi-hazard vulnerabilities [61–66]. However, comparative analyses of depopulation and vernacular architecture on a larger scale continue to be scarce [67]. This research aims to establish an understanding of how a negative demographic evolution has influenced vernacular architecture. The main aim is to obtain statistical data for the vulnerability factors threatening vernacular architecture in depopulated territories on a macroscopic

level. This entails the creation of a map illustrating the level of physical risk faced by vernacular architecture on a territorial scale and determining the urgency for intervention. To do so, a general analysis of a series of case studies is used as the starting point for the creation of a database recording the levels of physical vulnerability of vernacular architecture in individual municipalities. The specific objectives are to offer a general overview of vernacular architecture in the territories affected by depopulation, to identify the risk to vernacular architecture of the individual case studies, to evaluate vulnerability factors in relation to the depopulation level of individual municipalities, and to draw up maps for the level of risk identified.

The new methodology analysis presented in this study quantifies the risk to vernacular architecture in a large number of case studies in the regions affected by depopulation. The analysis also establishes the current condition of this vernacular architecture after over seventy years subjected to the phenomenon of depopulation, and makes it possible to determine whether the phenomenon is a determining factor in the generation of high levels of risk. Thus, the statistical results obtained for the 180 municipalities facilitate the creation of strategies and guidelines tailored to the current risk levels of individual municipalities. Considering the trend of rural depopulation in the world, which is especially noticeable in Europe [3], the methodology and results obtained in this study on the risks, threats and factors of resilience of vernacular architecture are applicable to other regions affected by the phenomenon of depopulation. Moreover, they might also help prevent the loss of vernacular architecture in these places where the decline in rural population is likely to worsen in the next few years.

## 2. Materials and Methods

### 2.1. Study Area

The study area, located in the region of Comunidad Valenciana (Spain), covers 180 municipalities from a total of 542 in the region (Figure 1). For the selection of case studies, it was considered necessary to add municipalities included within two main sources of information on depopulation in Comunidad Valenciana. One of these is the Anti-depopulation Agenda for Valencia (AVANT) [68], which includes 172 municipalities funded by the Municipal Cooperation Fund for the Fight against Depopulation. The other is the Valencia Institute of Statistics (IVE) [69]. According to demographic indicators of depopulation, 173 municipalities in Comunidad Valenciana are at moderate, high or very high risk of depopulation.

As both institutions use different criteria to assess depopulation, and this study does not aim to ascertain which of these are correct, for the selection of case studies, municipalities have been combined from the Anti-depopulation Agenda for Valencia (AVANT) and the Valencia Institute of Statistics (IVE). This combination has resulted in a list of 182 municipalities affected by depopulation in Comunidad Valenciana. Following analysis of the 182 municipalities identified by both institutions, two, the municipalities of Benagéber and Domeño, were ruled out as case studies. Both were transferred and rebuilt after the creation of the Reservoir of Benagéber in 1950 and the Reservoir of Loriguilla in 1979, respectively. The creation of these reservoirs entailed the loss of centuries-worth of vernacular architecture in these former villages. For this reason, excluding the municipalities of Benagéber and Domeño from the analysis is consistent with the research aim of analyzing the effects of depopulation on vernacular architecture, as no traces of their vernacular architecture are conserved.

In 1900, the total population of the 180 case studies was 230,117 inhabitants. In 1910, this amount totaled 234,005 inhabitants. This last census was taken the year that the number of inhabitants peaked. From that point on, the phenomenon of depopulation was found in the case studies. In 1950, the total had fallen to 202,064 inhabitants, a 14% reduction in population. Population loss became more intense from the 1950s to 2022. From 1950 to 1960, this variation was 12%; from 1961 to 1970, it was 22%; from 1971 to 1980, it was 17%; from 1981 to 1990, it was 11%; from 1991 to the turn of the century, it varied by 6%; from the 2001 census to 2011, the population level remained the same; and

between 2011 and the 2022 census, the population decreased by a further 9%. Overall, since the phenomenon of depopulation intensified in 1950, the population of the case studies went from 202,064 inhabitants to 85,911 inhabitants in 2022, the equivalent of a 57% loss in population [70] (Figure 2).

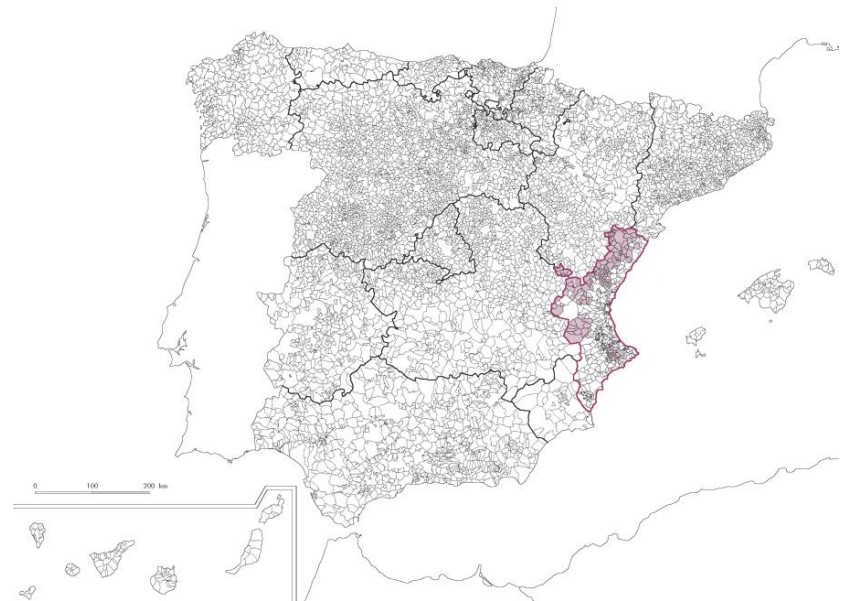

**Figure 1.** Map of Spain showing Comunidad Valenciana outlined and the 180 case studies shaded.

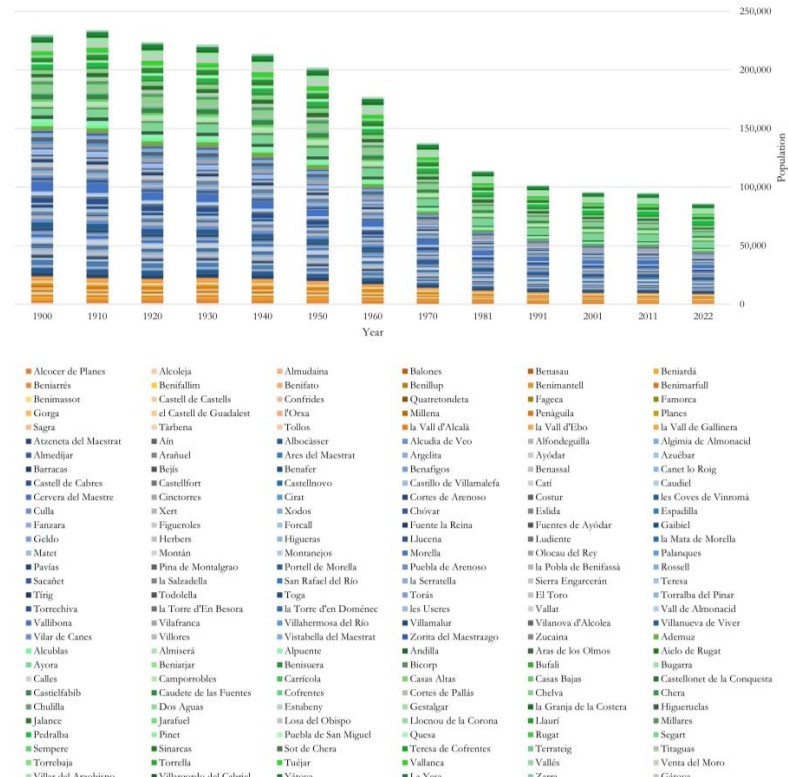

**Figure 2.** Changes in population in 1900–2022. Drawn up by the authors based on population censuses since 1842 obtained from the Spanish National Statistics Institute. According to the legend colors, the municipalities in orange correspond to the case studies in the Alicante province, the municipalities in blue and gray correspond to the case studies in the Castellón province and the municipalities in green correspond to the case studies of the Valencian province.

### 2.2. Classification of Case Studies for Analysis

Depopulation in itself is the main risk factor in this study. However, the analysis of different factors threatening vernacular architecture depending on the levels of depopulation observed in individual municipalities is proposed. Thanks to this, it can be ascertained whether the higher levels of depopulation result in a greater vulnerability of vernacular architecture or whether, in contrast, the vulnerability of vernacular architecture depends on the random nature of each factor. For this purpose, the analysis groups case studies by loss of population experienced between 1950 and 2022 (Figure 3). Five groups are created based on the population loss in the years mentioned:

- Group 1: population loss ≥ 80%: 26 case studies;
- Group 2: 60% ≤ population loss < 80%: 70 case studies;
- Group 3: 40% ≤ population loss < 60%: 53 case studies;
- Group 4: 20% ≤ population loss < 40%: 18 case studies;
- Group 5: 20% > population loss: 13 case studies.

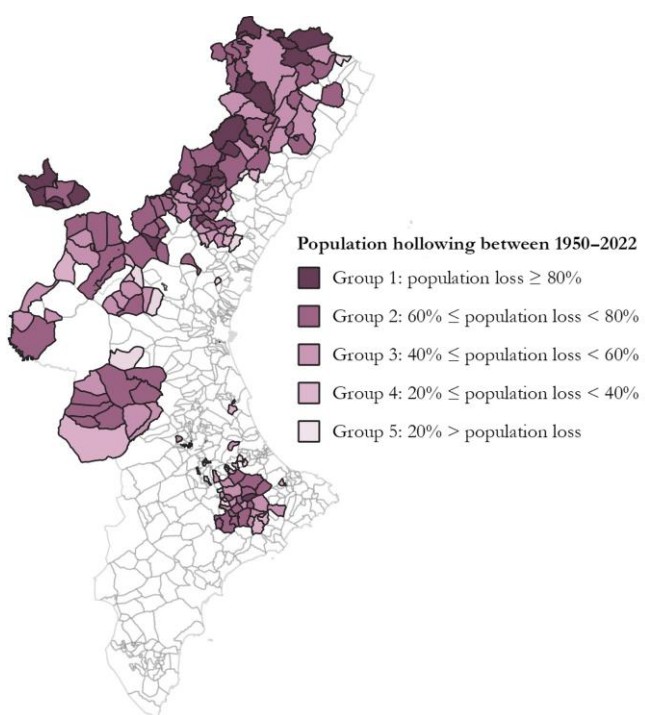

**Figure 3.** Maps showing case study classification according to depopulation levels between 1950 and 2022.

### 2.3. Physical Vulnerability Indicators

The assessment of the level of physical risk to vernacular architecture depends on the indicators that reflect the current condition of dwellings in the selected territories. In the introduction, the risks and hazards identified by researchers have been classified and translated into different vulnerability factors affecting vernacular architecture in depopulated territories. Within the framework of physical and material risks, a series of threats due to depopulation has been detected, including abandonment and progressive ruin, the neglect of traditional techniques, the poor state of rural housing, unsuitable practices and the introduction of incompatible materials, and the lack of sensitivity and appreciation on the part of owners and new generations. In order to analyze the current condition of vernacular dwellings in the territories affected by depopulation on a macro scale, and due to the complexity and diversity of territories, these risks have resulted in three common indicators of physical vulnerability factors chosen for all case studies: the state of conservation, the type of use of the dwellings and the number of vernacular dwellings preserved, taking as a reference the data from the decade prior to the start of the depopulation process.

2.3.1. The Age of the Construction

Due to the risks to vernacular architecture associated with depopulation, abandonment [23,24], lack of recognition of the new generations [27,29–34], and unsuitable regulations, which often leads to demolition, imitation, vulgarization and homogenization of vernacular architecture [26,27,30,32,44,45,47], have led to the loss of a large number of vernacular constructions over the years. Between 1910 and 1950, the population decreased by 14% (see Figure 2). However, this is barely reflected in the number of households, which decreased by only 3% in the same period [70]. This is decisive for establishing 1950 as the reference date for the data selection of the research, given that an initial loss of 3% can be assumed without compromising the final results. Furthermore, it has been taken into account that industrialized materials and globalized building systems reached rural areas later than urban areas. In all likelihood, they were introduced after the process of intense emigration had already started. Consequently, in the case studies analyzed, the traditional way of building was still preserved until 1950; at which point, the effects of depopulation on vernacular architecture became apparent. Therefore, this research considers all housing built before 1950 to be vernacular, whereas dwellings built after this point are assumed to have been constructed using non-vernacular industrialized techniques and materials.

Dating the buildings makes it possible to ascertain the total number of vernacular dwellings in the case studies and establish the percentage of these in relation to the total number of dwellings in a given municipality. The results obtained show the representativeness of the vernacular constructions of a case study. Dating dwellings also enables comparison with the number of homes existing in 1950 (prior to the start of the depopulation phenomenon), and provides the number of samples lost following depopulation.

2.3.2. The Use of the Dwelling

In addition to the abandonment of dwellings [23,24], depopulation leads to changes in use by the new generations who inherit and inhabit these buildings [27,29–34]. In this respect, migration impacts both the population remaining in these increasingly depopulated places and the inhabitants moving to these new places. This leads to a loss of identity and sociocultural expression of their roots. It is therefore common for the migrant population to never fully leave, maintaining their links to their place of origin [71]. Urban inhabitants not only maintain their cultural and family connections, but in most places, they also conserve family properties such as dwellings, rural constructions and crop fields. The second residence thus becomes a second home due to the heritage ties rooted in the past [72].

In this respect, the type of use of vernacular dwellings can affect the conservation of vernacular architecture. The abandonment of vernacular dwellings entails greater physical risks such as lack of maintenance, which can lead to more serious pathologies. In contrast, regular use of the vernacular dwelling ensures daily maintenance, as well as an interior layout that can maintain the traditional layout of a vernacular dwelling to a greater extent, thus ensuring an interior preservation of traditional spaces and construction techniques. However, a seasonal use of the vernacular dwelling may involve more complex adaptation processes, as it does not require the same layout as a traditional dwelling. For generation after generation, the families of the migrant population have continued to grow, so that seasonal visitors from a single family are far more numerous than when their relatives emigrated, while the number of family properties in the municipality has remained the same. This may entail the need to increase the number of rooms in order to maximize the accommodation capacity for these homes. This could also involve extensions and modifications that might present a risk to the conservation of the vernacular architecture.

Consequently, the type of use given to the dwelling influences the conservation of vernacular architecture, irrespective of whether it is a primary home (regularly occupied), a secondary home (occasionally occupied) or empty (permanently unoccupied). Thanks to this, an analysis can be carried out depending on the type of use of the dwelling in the case studies, making it possible to obtain the number of abandoned dwellings.

### 2.3.3. The State of Conservation of the Dwelling

Abandonment [23,24], changes in use and the lack of recognition by new generations [27,29–34], along with unsuitable regulation [26,27,30,44–47] of vernacular architecture, often lead to unsuitable alterations and a lack of maintenance of the dwellings. This situation can evolve, causing pathologies, which, if untreated, can bring about more serious issues, jeopardizing the conservation of vernacular architecture. Therefore, the state of conservation of the dwelling is a relevant indicator when establishing the state of conservation of the vernacular architecture of the case studies, defined as good (when the building displays none of the circumstances defined for states of ruin and poor or deficient states), deficient (when the building displays any of the following circumstances: when roof drainage or sewage are in poor condition, damp in the lower part of the building or infiltrations in roofs or ceilings), poor (when one or more of the following occur in the building: large cracks or bulging in some of the façades, subsidence or lack of horizontality in ceilings or floors or the foundations of the building) or ruinous (when any of the following situations are present in the building: it is shored up, or an official declaration of ruin is being processed or has been confirmed).

This study considers any buildings in ruins, or in a poor or deficient condition to be a possible risk, and from this point on, they are simply unified using the adjective poor (as opposed to good). This allows analysis to be carried out according to the condition of the dwelling in the case studies, also providing the number of dwellings in conditions deemed uninhabitable or insalubrious.

In order to ascertain the current condition of vernacular architecture, the indicators of physical vulnerability (year of construction, type of use and state of conservation) can be combined so that the current condition of vernacular dwellings in the case studies can be obtained.

### 2.4. Data and Methodology

This research is based on the statistical quantitative analysis of the physical vulnerability factors threatening vernacular architecture. The National Statistics Institute publishes the results of the Population and Housing Census every ten years. It is through these censuses that the physical vulnerability indicators selected for the research have been analyzed. Using the reference data from decades prior to the start of the depopulation process, the vulnerability factors (the year of construction, the type of use and state of conservation) were obtained and compared in relation to the number of dwellings in each municipality, aiming to understand the evolution of vernacular architecture in the case studies. Thanks to an initial compilation of historical and current information on the case studies, data on the amount of vernacular architecture affected by depopulation and the level of risk currently affecting this architecture can be observed.

Historical demographic data for the period between 1842 and 1981 were obtained from the Intercensal Population Census of the Spanish National Statistics Institute [70]. Current data on demographics and dwelling were obtained from the most recent Population and Housing Censuses [73], for the years 1981, 1991, 2001, 2011 and 2021. The most complete and up-to-date information available for these vulnerability factors is that found in the 2011 census, as the 2021 census is still being processed and has not yet been completely published. Therefore, the 2011 census was used to obtain data on the type of use, year of construction and state of conservation of the dwelling.

After defining the criteria for the search and collection of information, this was organized in order to create a clear cohesive database. This database, with independent values, allows the data to be viewed and managed for the different case studies, as well as providing clear objective statistical or comparative data. Excel spreadsheets are used to handle the data obtained following the information search. This is due to the ease of creating a personalized database with variables in both rows and columns. This software allows data to be handled, as well as to be filtered by category, creating tables and graphs

to interpret these data, as well as the export of information to other programs used for the creation of risk maps.

The case study classification includes the necessary information to identify the individual case studies and filter by category for subsequent analysis. This information includes the following:

- INE code: This is the unique five-digit number assigned to each municipality, establishing the province and number of each individual municipality. Thanks to this code, data from different sources can be linked, as the system is used for the identification of municipalities throughout Spain.
- The name of the municipality, region and province it is located in. This allows filtering by geographical zone and the comparison of vulnerability factors according to individual case study locations.
- The data source on depopulation, whether it is part of Agenda AVANT, and the level of risk according to IVE. The selection of case studies based on these two data sources enables the subsequent analysis of vernacular architecture to ascertain if one source is more vulnerable than the other.
- The group by demographic data. The demographic analysis of case studies establishes five groups based on the population loss undergone from 1950 to 2022. This makes it possible to analyze the physical vulnerability factors according to depopulation levels and to establish whether the level of risk of vernacular architecture depends on depopulation or is a random factor.
- Type of dwelling: primary, secondary or empty.
- State of the building: good, deficient, poor or ruinous.
- Year of construction of the building.

After inputting the information into the database, a statistical analysis was carried out on individual vulnerability factors. Statistical data were obtained individually for each physical vulnerability factor for the different case studies. Thanks to this, numerical data were obtained on the current condition of vernacular architecture, the number of samples conserved in each municipality and its current state of conservation. In addition, case studies grouped according to the total population loss between 1950 and 2022 made it possible to ascertain whether the depopulation level affects the vulnerability of vernacular architecture.

The definition of each factor and the identification of the risk for each case study through tables, statistics and graphs make up a full study. However, these representation systems are complex due to the large number of case studies. In order to guarantee the data are easily understood, the creation of risk maps showing the risk level to the vernacular architecture of individual case studies was proposed. This research relied on the fundamental tool of GIS (Geographic Information System) for its development. This work system enables the collection, management and analysis of spatially located data, organizing information by layers and creating maps for the analysis of information in order to identify problems, understand trends and establish priorities. This study relied on the use of QGIS, a free open-code GIS program [74]. The potential of this tool, which even allows the creation of new data, lies in the cross-referencing of different characteristics. In this case, the data obtained previously from tables and graphs regarding the different vulnerability factors and risk levels to vernacular architecture in individual case studies are entered into the QGIS software, version QGIS Desktop 3.16.13. This is made possible by the fact that individual case studies have been assigned the same INE code, both in official IDEV maps [75] and in statistical data from INE and IVE.

This statistical analysis of vernacular architecture develops a useful analytical tool to lay the foundations for the analysis of specific case studies, establishing a context for the situation of vernacular architecture, as well as a classification by the degree of urgency for the risks to be addressed. Comparison with other case studies allows the public administration to determine priority measures according to the level of risk. In turn, it is possible to previously ascertain the main risk factors in a municipality beforehand, in

order to guide the fieldwork aimed at obtaining a better understanding of the situation of the vernacular architecture of a case study. Furthermore, this allows specific strategies to be proposed for the individual municipalities or geographical areas based on the factors displaying the greatest vulnerability.

### 2.5. Calculation of Vulnerability Factor and Risk Level

2.5.1. Vulnerability Factor: Variation in Dwellings Built before 1950

The first category, thanks to which the current condition of vernacular architecture can be established, is the age of the dwellings of a municipality. Since 1991, Population and Housing Censuses have taken into account the number of dwellings in individual municipalities according to their year of construction. By contrasting this piece of data with the number of homes recorded in 1950, the loss of vernacular dwellings following the phenomenon of depopulation can be determined.

The formula for percentage variation between the number of homes in 1950 and the dwellings built before 1950 is used in order to obtain the results of this first vulnerability factor. This formula considers the analysis groups defined in order to establish the level of risk to the census population in relation to the level of depopulation (k = group 1, group 2, group 3, group 4, group 5):

$$V(k) = 100 * \sum_k \frac{\left(x_f - x_i\right)}{(x_i)}$$

where

$V(k)$ = Variation in dwellings built before 1950 according to analysis groups;
$x_f$ = Value of dwellings built before 1950 in the 2011 census;
$x_i$ = Value of homes in 1950;
$k$ = Analysis groups.

The vulnerability of vernacular architecture, in terms of the variation in the number of homes in 1950 in relation to the dwellings of the 2011 census built before 1950, speaks for itself. The greater the percentage variation, the greater the risk affecting the vernacular architecture of a given municipality. Moreover, this vulnerability factor is the only factor in this research that has no margin for improvement, as once vernacular architecture is lost, it cannot be recovered. In order to establish the risk level that this vulnerability factor poses to vernacular architecture, a risk level is assigned according to the variation in the number of dwellings in 1950 in relation to case study dwellings built prior to 1950 and appearing in the 2011 census:

| | |
|---|---|
| 1 = Null: | percentage variation $\geq -10\%$; |
| 2 = Medium: | $-10\% >$ percentage variation $\geq -25\%$; |
| 3 = High: | $-25\% >$ percentage variation $\geq -50\%$; |
| 4 = Critical: | percentage variation $< -50\%$. |

2.5.2. Vulnerability Factor: Empty Dwellings Built before 1950

One of the threats linked to depopulation is the abandonment of the properties. This entails increased risks such as lack of maintenance and conservation of architecture, with the subsequent decay that can speed up the process of it being declared in ruins. The second category used to establish the current condition of vernacular architecture is the type of use given to the dwellings in a municipality. Since 1991, censuses for Population and Housing consider the number of dwellings in municipalities according to whether their use is primary or secondary or whether they are empty.

The negative consequences of the abandonment of dwellings mean that the percentage of empty dwellings built between 1950 is a vulnerability factor for vernacular architecture in the case studies. In order to obtain the results of this analyzed vulnerability factor, the percentage of the number of empty dwellings built before 1950 is analyzed in relation to

the total of dwellings built before 1950. This formula takes into account the analysis groups defined in order to obtain the risk level for the population in the census in relation to the level of depopulation (k = group 1, group 2, group 3, group 4, group 5):

$$P(k) = 100 * \sum_k \frac{(x_v)}{(x_t)}$$

where

$P(k)$ = Percentage of empty dwellings according to the analysis group;
$x_v$ = Value of empty dwellings built before 1950 and appearing in the 2011 census;
$x_t$ = Total value of dwellings built before 1950 and appearing in the 2011 census;
$k$ = Analysis group.

The risk entailed by the abandonment of vernacular architecture, both in social (loss of cultural identity of a place) and physical terms (loss of traditional knowledge and techniques), shows the need to calculate the risk level entailed for each case study. Therefore, the percentage of empty dwellings in relation to the total number of dwellings built before 1950 is considered a vulnerability factor for vernacular architecture. The higher the percentage of empty dwellings, the higher the level of risk to vernacular architecture. In order to determine the risk level of this vulnerability factor for vernacular architecture, a risk level is assigned to the percentage of the number of empty case study dwellings in relation to the total number of dwellings built before 1950 and appearing in the 2011 census:

1 = Null:                 percentage $\leq$ 10%;
2 = Medium:             10% < percentage $\leq$ 20%;
3 = High:                 20% < percentage $\leq$ 30%;
4 = Critical:             percentage > 30%.

2.5.3. Vulnerability Factor: Dwellings in Poor Condition Built before 1950

The third category, which establishes the current condition of vernacular architecture, corresponds to the state of conservation of the dwellings in a municipality. Since 1991, the Censuses of Population and Housing have taken into account the number of dwellings in individual municipalities depending on whether their condition is good, deficient, poor or ruinous. Due to the importance of the state of dwelling as a vulnerability factor to identify the current condition of vernacular dwellings, the condition of dwellings built before 1950 is analyzed according to the analysis groups created. In order to obtain the results of this analyzed vulnerability factor, the percentage of the number of dwellings in poor condition built before 1950 is calculated in relation to the total number of dwellings built before 1950. This formula considers the analysis groups defined in order to obtain the level of risk of the census population in relation to the level of depopulation (k = group 1, group 2, group 3, group 4, group 5):

$$P(k) = 100 * \sum_k \frac{(x_r + x_p + x_d)}{(x_t)}$$

where

$P(k)$ = Percentage of dwellings in poor condition according to the analysis group;
$x_r$ = Value of dwellings in a state of ruin built before 1950 in the 2011 census;
$x_p$ = Value of dwellings in poor condition built before 1950 in the 2011 census;
$x_d$ = Value of dwellings in deficient condition built before 1950 in the 2011 census;
$x_t$ = Total value of dwellings built before 1950 in the 2011 census;
$k$ = Analysis group.

By finding out the state of conservation of the architecture built before 1950, it is possible to ascertain the current condition of vernacular architecture and identify the potential risk level if not suitably conserved. The higher the percentage of dwellings in

poor condition, the greater the level of risk to vernacular architecture. In order to determine the risk level of this vulnerability factor in relation to vernacular architecture, a risk level is assigned to the number of dwellings in poor condition in relation to the total case study dwellings built before 1950 and appearing in the 2011 census:

1 = Null:                              percentage $\leq$ 10%;
2 = Medium:                       10% < percentage $\leq$ 20%;
3 = High:                             20% < percentage $\leq$ 30%;
4 = Critical:                         percentage > 30%.

2.5.4. Vulnerability Factor: Empty Dwellings in Poor Condition Built before 1950

The last vulnerability factor directly affecting the physical conservation of vernacular architecture combines all the factors above. Thanks to this factor, it is possible to obtain the statistical number of dwellings built before 1950, and those that are currently empty or in a poor state of conservation. The combination of the three factors results in a rather unique situation that can manifest the imminent risk of loss of vernacular architecture in a municipality. In order to obtain the results of this analyzed vulnerability factor, the percentage of the number of empty dwellings in poor condition built before 1950 is calculated. This formula takes into account the analysis groups defined to ascertain the level of risk of the census population in relation to the level of depopulation (k = group 1, group 2, group 3, group 4, group 5):

$$P(k) = 100 * \sum_k \frac{(x_v)}{(x_t)}$$

where

$P(k)$ = Percentage of empty dwellings according to analysis groups;
$x_v$ = Value of empty dwellings in poor condition built before 1950 in the 2011 census;
$x_t$ = Total value of dwellings built before 1950 in the 2011 census;
$k$ = Analysis group.

The combination of all the vulnerability factors affecting the dwelling entails an added risk to a dwelling built prior to 1950. If urgent action is not carried out on the entire abandoned vernacular dwelling, in a poor state of conservation, this can lead to a greater loss of vernacular architecture in the case studies. Therefore, knowing the number of dwellings that are empty and in poor condition built before 1950 allows an understanding of the current situation of vernacular architecture, manifesting the risk to which it is subject if not properly conserved. In order to calculate the risk of individual case studies, the percentage of empty dwellings in poor condition in relation to the total number of dwellings built before 1950 is used. The higher the percentage of empty dwellings in poor condition, the greater the risk level to vernacular architecture. In order to establish the risk level of this vulnerability factor for vernacular architecture, a risk level is assigned to the percentage of the number of empty dwellings in poor condition in relation to the total case study dwellings built before 1950 and appearing in the 2011 census:

1 = Null:                              percentage $\leq$ 5%;
2 = Medium:                       5% < percentage $\leq$ 10%;
3 = High:                             10% < percentage $\leq$ 20%;
4 = Critical:                         percentage > 20%.

## 3. Results

### 3.1. Variation in Dwellings Built before 1950

The results obtained show a significant loss between the total of dwellings in 1950 and the number of dwellings built before 1950 appearing in the 2011 census. In total, there were 63,010 dwellings in 1950. In 2011, there were 37,360 dwellings built before 1950. This means that over the 60-year period since the depopulation phenomenon began, 25,650 vernacular

dwellings have been destroyed, which equals a loss of 41% of vernacular dwellings in the case studies (Figure 4).

| | Census 1950 Dwellings | Census 2011 Dwellings built before 1950 | Percentage variation |
|---|---|---|---|
| Group 1 | 6839 | 4075 | −40% |
| Group 2 | 25,644 | 15,505 | −40% |
| Group 3 | 20,196 | 12,055 | −40% |
| Group 4 | 5852 | 3380 | −42% |
| Group 5 | 4479 | 2345 | −48% |
| Total | 63,010 | 37,360 | −41% |

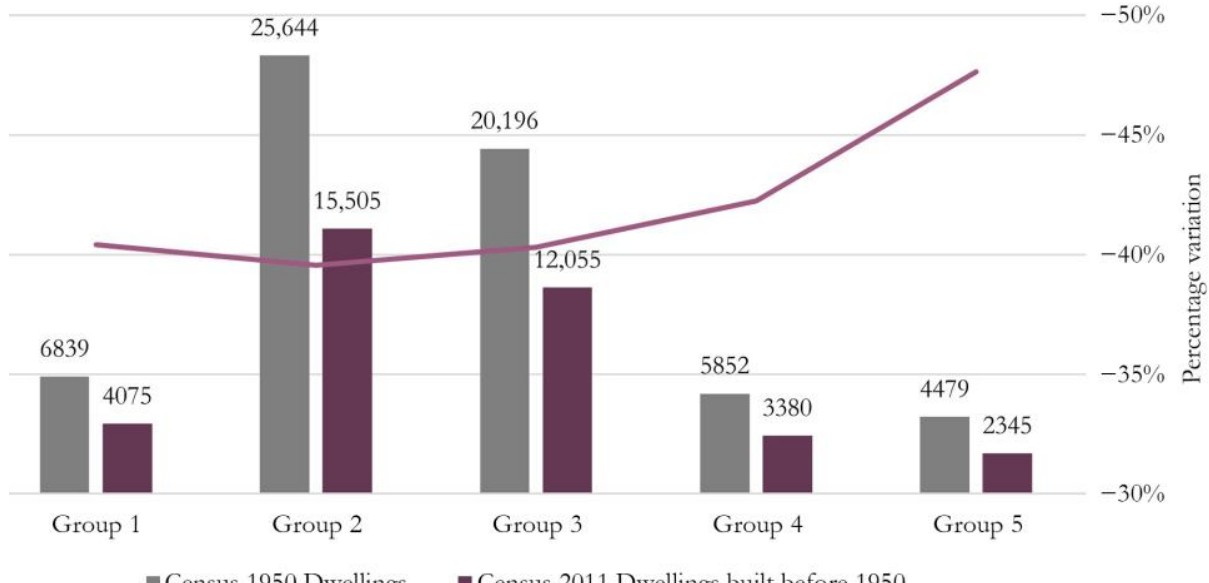

**Figure 4.** Variation in the number of dwellings in 1950 in relation to the dwellings built before 1950 and appearing in the 2011 census for the case studies classified according to the groups sorted by depopulation level (see Figure 3).

The same data, analyzed in relation to groups assigned by depopulation level, show a clear correlation between depopulation level and the loss of vernacular dwellings. The first three groups, which are the ones with the highest level of depopulation, maintain a constant loss of approximately 40%. However, the fourth and fifth groups, with the lowest emigration levels among the population, are case studies displaying a greater loss of vernacular dwellings. In this case, more intense emigration has resulted in a greater level of conservation of vernacular architecture.

The current situation of vernacular architecture for the case studies shows that almost half of the vernacular dwellings have been lost, potentially resulting in a loss of local identity and greater difficulty in the study of local construction techniques. In itself, depopulation leads to a lack of transmission of techniques, knowledge and values from one generation to another. As a result, in many municipalities affected by this phenomenon, it has not been possible to transmit vernacular architecture from one generation to another. Therefore, in most cases, the loss of vernacular buildings entails a material loss of great cultural and educational value for traditional societies. In short, 38 case studies show a null risk level, 28 cases show a medium risk level, 52 case studies show a high risk level, and 62 case studies show a critical risk level (Figure 5).

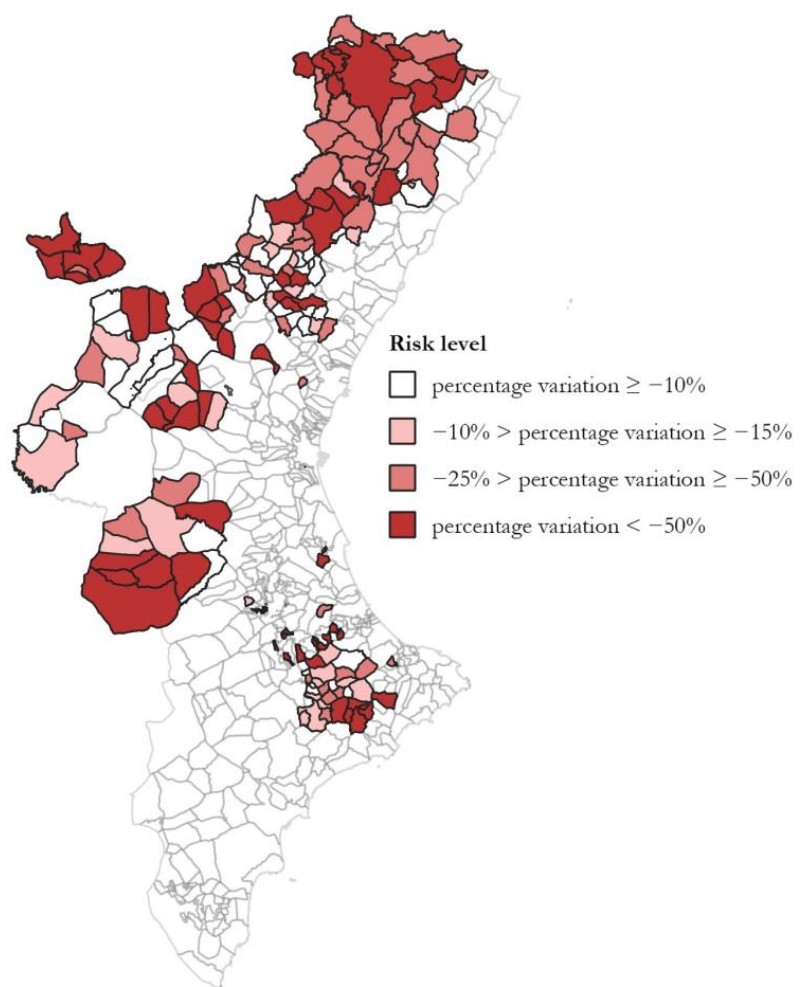

**Figure 5.** Risk level according to the variation in the number of dwellings in 1950 in relation to the dwellings built before 1950 and appearing in the 2011 census.

### 3.2. Empty Dwellings Built before 1950

From 1950 to 2011, in addition to the irrecoverable loss of 25,650 vernacular dwellings, depopulation has meant that, of the 37,360 dwellings conserved, 21% of those built before 1950 currently lie empty (Figure 6). The results obtained for the empty dwellings built before 1950 show the current condition of vernacular architecture. The higher the level of depopulation, the lower the percentage of empty dwellings built before 1950. Equally, the lower the depopulation level, the greater the number of empty case study dwellings built before 1950, and these are the case studies that show the greatest loss of vernacular architecture during depopulation (Figure 7).

This vulnerability factor must be understood from a prior condition, where conserved vernacular architecture already represents, on average, only 60% of what it once was, and accounts for only 39% of all the dwellings in the municipality. Therefore, a minimal percentage of abandoned dwellings may lead to a loss in vernacular architecture that the case studies cannot withstand. If this loss increases, the ratio between dwellings built with industrialized techniques and materials may result in only one of every three dwellings being vernacular. In short, 47 case studies display null risk, 56 case studies display a medium risk level, 31 case studies display a high risk level, and 46 case studies display a critical risk level (Figure 8). Figure 9 shows an example of an empty vernacular dwelling, in which most of the preserved traditional techniques can be detected.

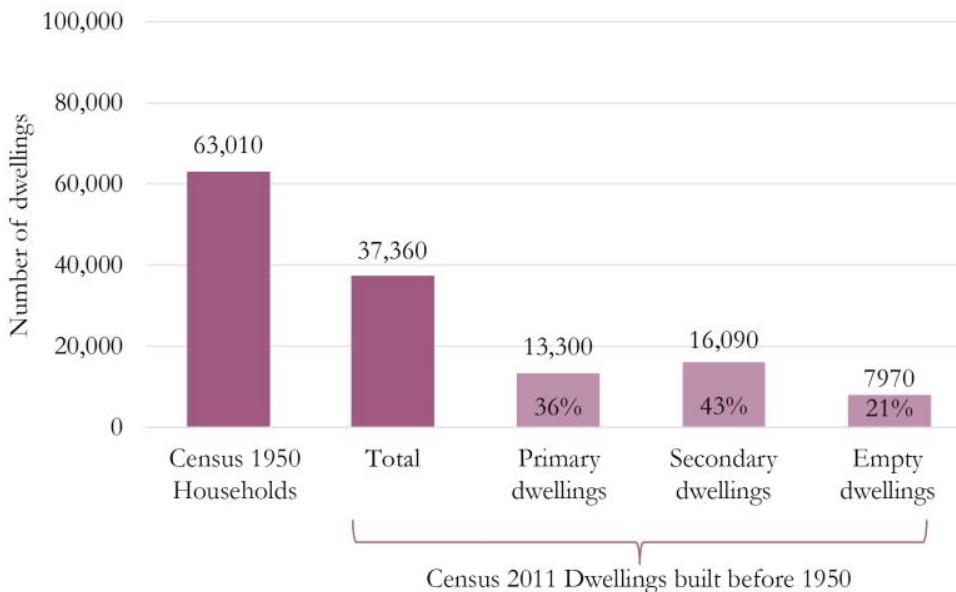

**Figure 6.** Number of households in 1950 and number of dwellings built before 1950, according to the type of use, in relation to the Population and Housing Census for 2011.

| | Empty dwellings built before 1950 | Total dwellings built before 1950 | Percentage of empty dwellings as a percentage of total dwellings built before 1950 | Census 1950 Dwellings | Census 2011 Total dwellings built before 1950 | Percentage loss between households in 1950 and dwellings built before 1950 in the 2011 Census |
|---|---|---|---|---|---|---|
| Group 1 | 545 | 4075 | 13% | 6839 | 4075 | 40% |
| Group 2 | 2700 | 15,505 | 17% | 25,644 | 15,505 | 40% |
| Group 3 | 2795 | 12,055 | 23% | 20,196 | 12,055 | 40% |
| Group 4 | 1085 | 3380 | 32% | 5852 | 3380 | 42% |
| Group 5 | 845 | 2345 | 36% | 4479 | 2345 | 48% |
| Total | 7970 | 37,360 | 21% | 63,010 | 37,360 | 41% |

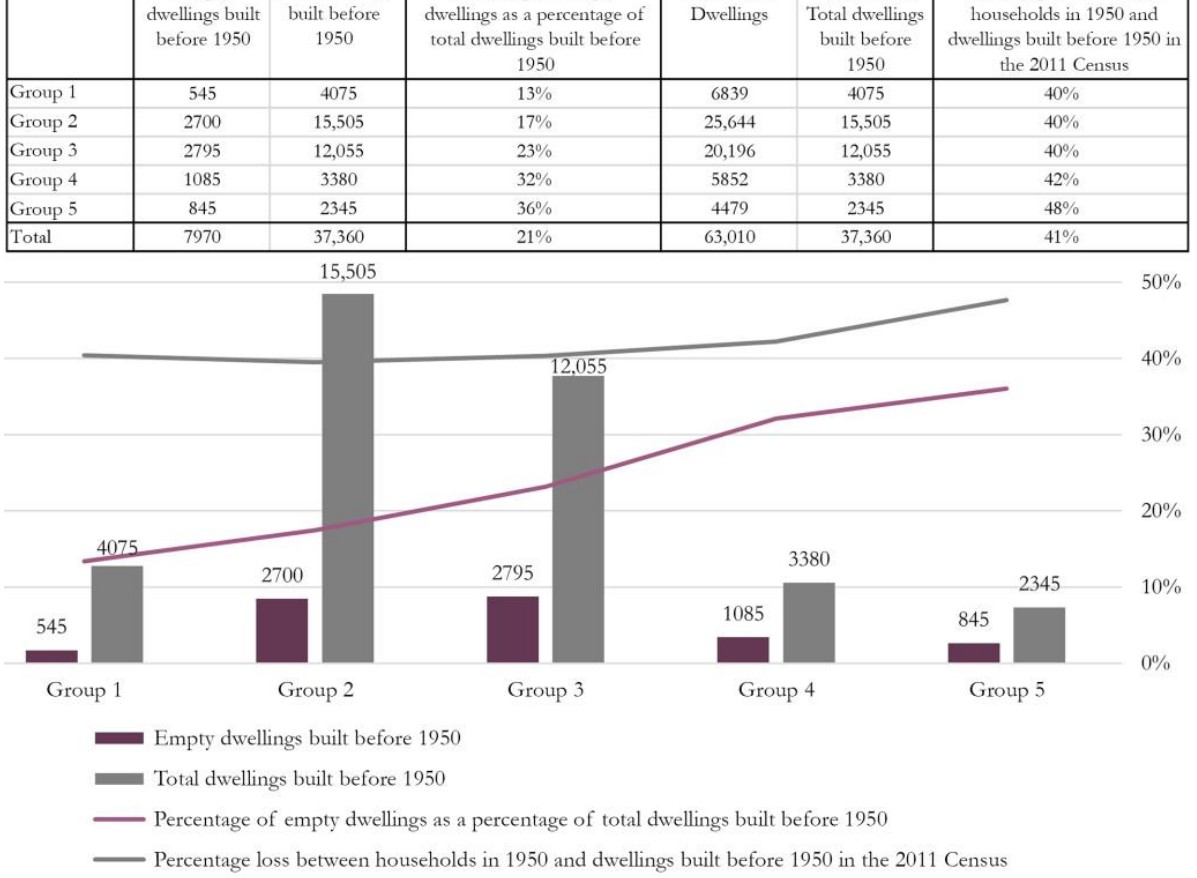

**Figure 7.** Percentage of the number of empty dwellings in relation to the total dwellings built before 1950 and appearing in the 2011 census from the case studies classified by groups assigned according to depopulation level (see Figure 3).

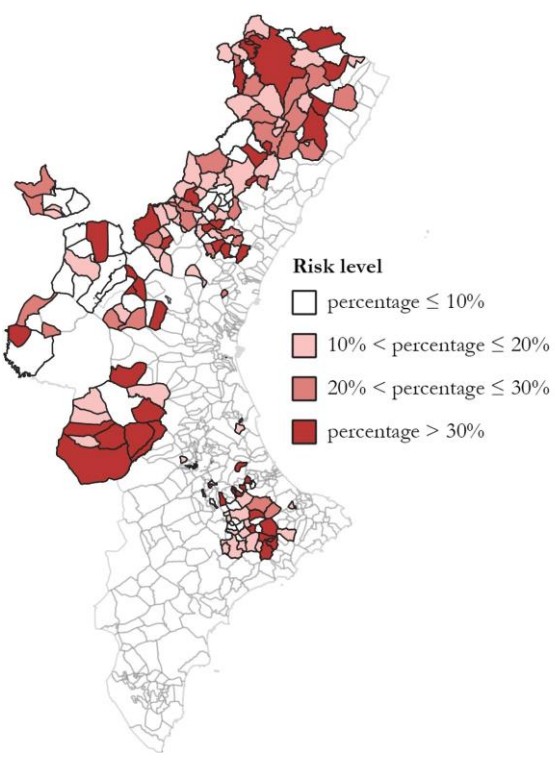

**Figure 8.** Risk levels according to the percentage of the number of empty dwellings in relation to the total dwellings built before 1950 and included in the 2011 census.

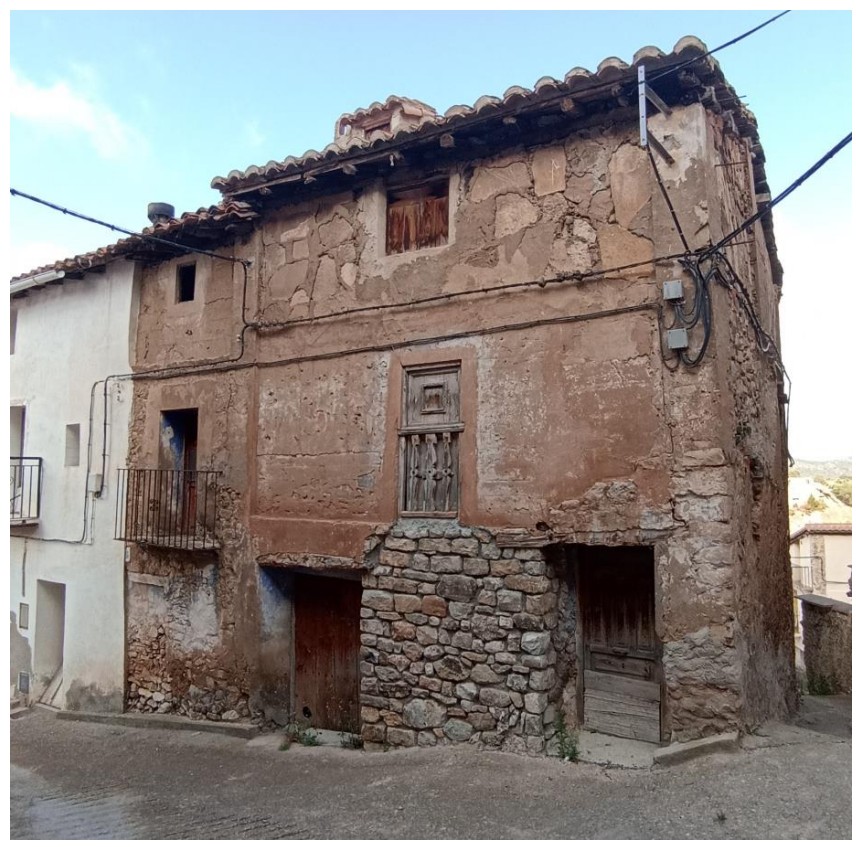

**Figure 9.** Empty vernacular dwelling on Calle de la Plaza, Puebla de San Miguel, 2022. Author: Eva Tortajada.

### 3.3. Dwellings in Poor Condition Built before 1950

In order to ascertain the condition of the vernacular dwellings of case studies, the information obtained from INE has been filtered once more, this time according to year of construction. According to the data, 28% of dwellings built before 1950 are in poor condition (Figure 10). This means that of the 37,360 vernacular dwellings conserved, 10,425 have some sort of pathology. There may be different causes for this, but it must be remembered that 21% of the dwellings built before 1950 are abandoned, which may be linked to the poor state of conservation of vernacular dwellings.

| Census 2011 | Ruinous | Poor | Deficient | Good | Total | Percentage of dwellings in poor condition |
|---|---|---|---|---|---|---|
| Group 1 | 55 | 280 | 860 | 2880 | 4075 | 29% |
| Group 2 | 320 | 850 | 3105 | 11,230 | 15,505 | 28% |
| Group 3 | 175 | 530 | 2175 | 9175 | 12,055 | 24% |
| Group 4 | 60 | 295 | 610 | 2415 | 3380 | 29% |
| Group 5 | 150 | 200 | 760 | 1235 | 2345 | 47% |
| Total | 760 | 2155 | 7510 | 26,935 | 37,360 | 28% |

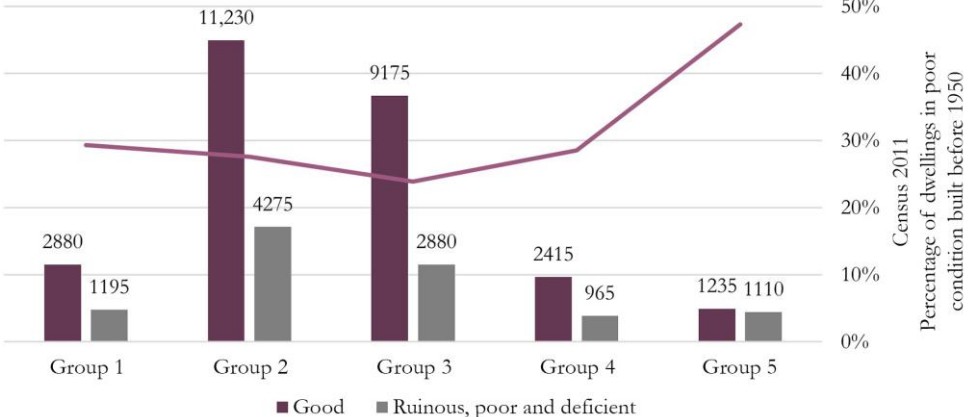

**Figure 10.** Number of dwellings in poor condition in relation to the total case study dwellings built before 1950 and appearing in the 2011 census, classified according to groups assigned by depopulation level (see Figure 3).

The results obtained from the analysis of vernacular dwellings in poor condition show a slight trend in the first three levels of depopulation. It can be observed that the greater the level of depopulation, the higher the percentage of dwellings built before 1950 in poor condition. These data are in contrast with the results obtained for the empty dwellings, as these are opposing trends. Equally, for the first three levels, statistics show that 27% of vernacular architecture is in poor condition, but only 18% of vernacular architecture is abandoned. This means that according to the statistics, there is more vernacular architecture in poor condition than empty dwellings (which means that there are dwellings with pathologies still in use).

Furthermore, the case study group showing the lowest level of depopulation is the group with the highest percentage of dwellings built before 1950 in poor condition. It should be noted that this group also shows the highest percentage of empty dwellings built before 1950, which means that these case studies are at great risk as almost half of the vernacular architecture conserved is in poor condition.

The percentage of dwellings in poor condition in relation to the total number of dwellings built before 1950 is used to calculate the risk level for individual case studies. The higher the percentage of dwellings in poor condition, the greater the level of risk to vernacular architecture. In short, 20 case studies display a null risk level, 52 case studies display a medium risk level, 30 case studies a high risk level, and 78 case studies a critical risk level (Figure 11). Figure 12 shows an example of a vernacular dwelling in a poor state

of conservation, in which the loss of the original coverings can be detected, as well as the absence of some windows and the presence of humidity in several areas.

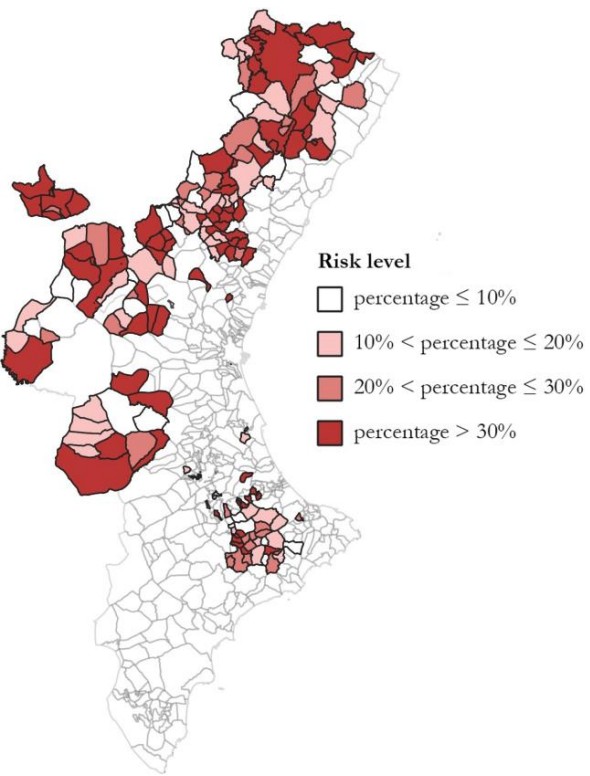

**Figure 11.** Risk level according to the percentage of the number of dwellings in poor condition in relation to the total number of dwellings built before 1950 appearing in the 2011 census.

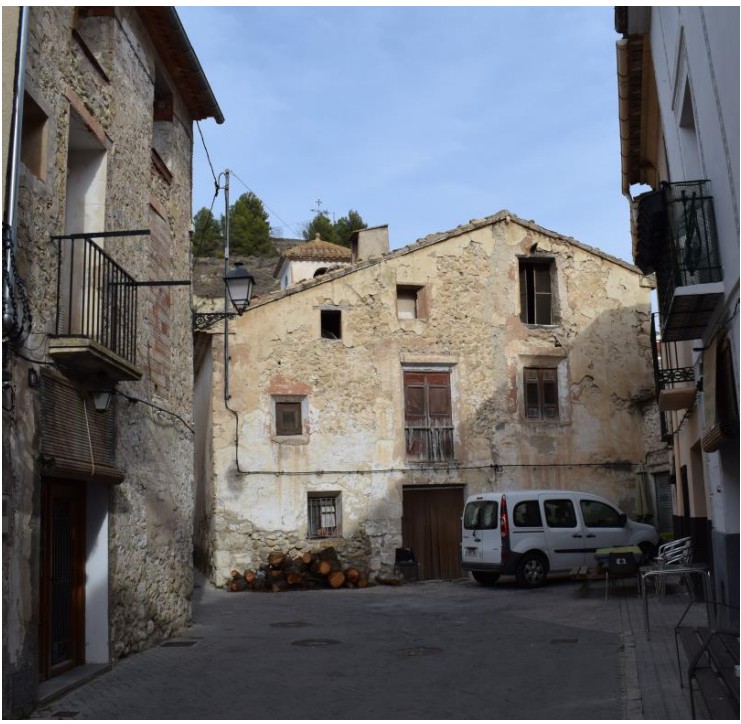

**Figure 12.** Vernacular dwelling in a poor state of conservation located on Calle de la Iglesia, Ares del Bosque, Benasau, 2024. Author: Eva Tortajada.

*3.4. Empty Dwellings in Poor Condition Built before 1950*

The results obtained from the situation of empty vernacular dwellings in poor condition show that the higher the level of depopulation, the lower the risk of loss of vernacular dwellings. The three groups with the highest levels of emigration since the 1950s show a percentage of empty housing in poor condition of around 9%. On the one hand, there is a positive aspect to this information, as small municipalities do not require major investments to improve the condition of dwellings since the number of dwellings in need of urgent intervention is low. On the other, if this situation is not resolved, the loss of a further 9% of vernacular dwellings constitutes a social, cultural, physical, economic and political risk, which should not be accepted. The municipalities that have suffered the least depopulation also have the highest percentage of empty dwellings in poor condition over the total number of dwellings built before 1950 and conserved (Figure 13). These case studies are the same ones that have already experienced greater loss in relation to households in 1950. In this situation, a higher percentage of empty dwellings in poor condition entails a high risk to the conservation of vernacular architecture and all associated values.

| Census 2011 | Empty, in poor state dwellings, built before 1950 | Total dwellings built before 1950 | Percentage of empty, in poor state dwellings, as a percentage of total dwellings built before 1950 |
|---|---|---|---|
| Group 1 | 360 | 4075 | 9% |
| Group 2 | 1305 | 15,505 | 8% |
| Group 3 | 1095 | 12,055 | 9% |
| Group 4 | 450 | 3380 | 13% |
| Group 5 | 505 | 2345 | 22% |
| Total | 3715 | 37,360 | 10% |

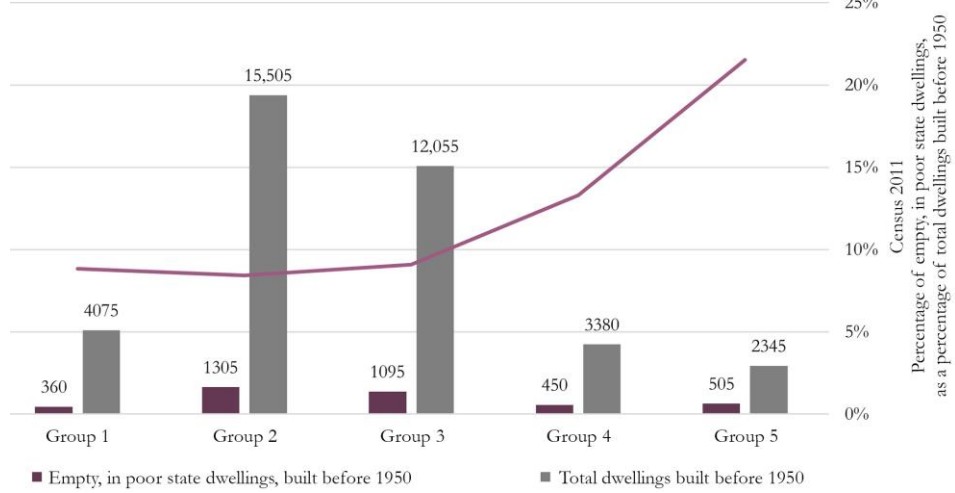

**Figure 13.** Number of empty dwellings in poor condition in relation to total case study dwellings built before 1950 and included in the 2011 census according to groups assigned by depopulation level.

In order to establish the risk level of this vulnerability factor for vernacular architecture, a risk level is assigned to the percentage of the number of empty dwellings in poor condition in relation to the total number of case study dwellings built before 1950 in the 2011 census. In short, 60 case studies display no risk level, 46 case studies display medium risk, 46 case studies are at high risk level, and 28 case studies are at critical risk level (Figure 14). Figure 15 shows an example of an empty vernacular dwelling in a poor state of conservation, in which the advanced state of deterioration could mean an imminent loss of this vernacular heritage.

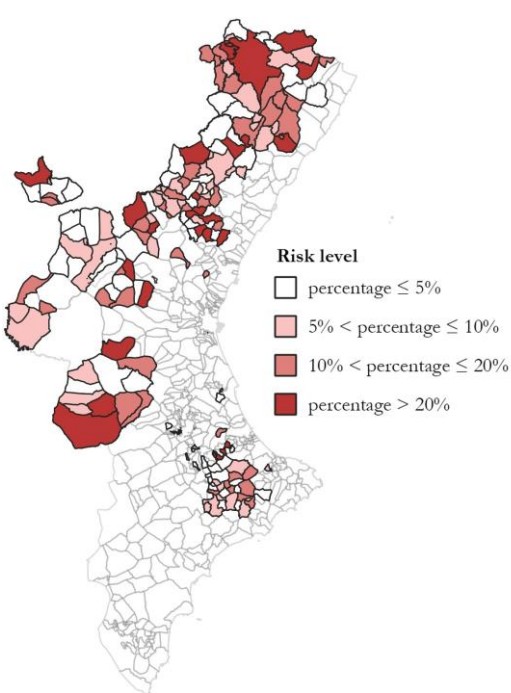

**Figure 14.** Risk level according to percentage of the number of empty dwellings in poor condition in relation to the total dwellings built before 1950 in the 2011 census.

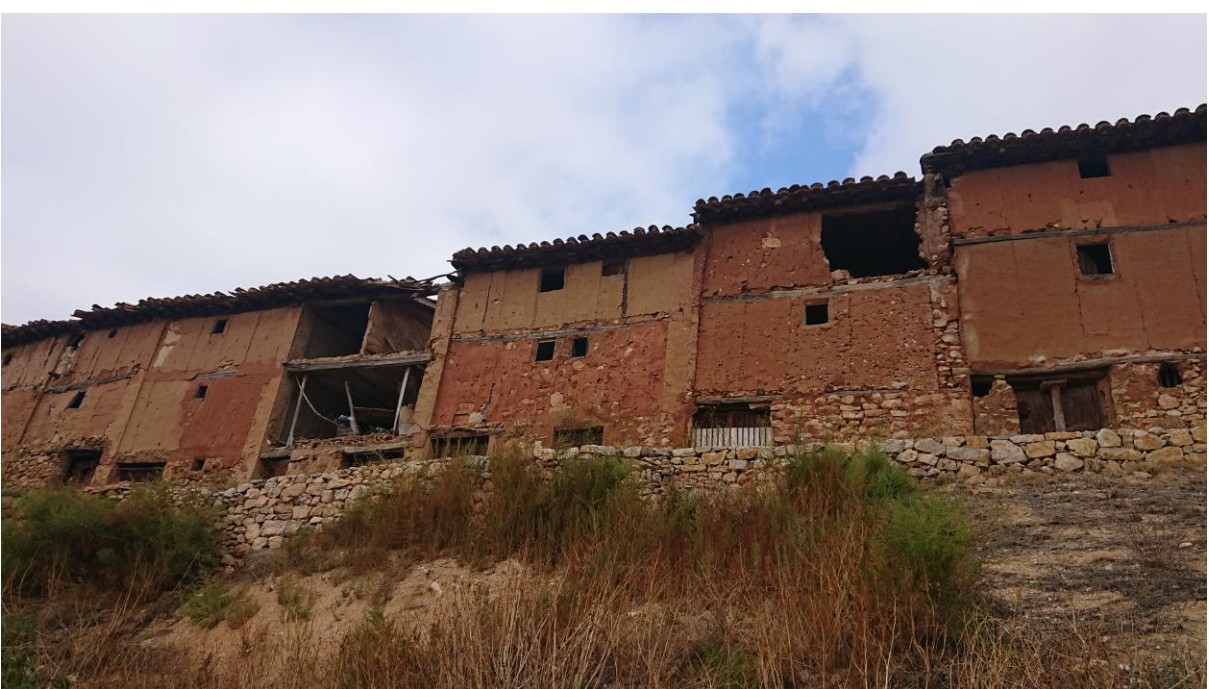

**Figure 15.** Empty vernacular dwellings in a poor state of conservation located on Calle del Barranquillo, Casas Bajas, 2021. Author: Eva Tortajada.

## 4. Discussion

In 1931, Torres Balbás stated that "it is necessary to go to the poorest and most remote places to find current and still lively manifestations of popular art" [23]. This quote referred to the municipalities far from urban societies, with little accessibility and no roads, in which it was possible to find vernacular architecture despite its slow disappearance due to industrial systems alien to traditions. Without referring to a specific place, any of the case

studies can be associated with the characteristics of the municipalities where Torres Balbás felt that vernacular architecture could still be seen.

Over almost a hundred years, the poorest and most remote places have now become the most depopulated municipalities. The statistical analysis of vulnerability factors shows a situation not dissimilar to that presented by Torres Balbás over 90 years ago. In this context, the municipalities that have experienced the most depopulation are the ones where vernacular architecture is best conserved. Thus, referring back to the warning by Torres Balbás, at present, in order to find vigorous representations of this vernacular art, it is necessary to go to the most depopulated places in Spain.

The places with the most surviving vernacular architecture are also the most depopulated ones. As a result, it is clear that vernacular architecture in these places is not spared from risk. The most notable risks connected with depopulation include physical risk due to abandonment and social risk due to the lack of recognition of the values of vernacular architecture. The combination of both risks constitutes a threat to vernacular architecture, often resulting in it being declared ruinous and subsequently demolished.

According to the results obtained, between 1950 and 2011, the vernacular architecture of the case studies decreased by 41%. Of the 37,360 dwellings conserved, 21% are abandoned and 28% are in a poor state of conservation. More specifically, 10% are empty and in a poor state of conservation. These 3715 dwellings are at risk of being declared as ruins and subsequently torn down. This, in turn, leads to vernacular architecture being replaced by new constructions, depleting its cultural richness, and homogenizing the character and identity of places [76].

The vulnerability factors analyzed in relation to the groups of analysis by level of depopulation suffered between 1950 and 2022 display similar tendencies. The greater the depopulation, the lower the physical risk to vernacular architecture. Specifically, the case studies showing the highest levels of depopulation have conserved a higher percentage of dwellings built before 1950, in addition to which, these municipalities have the lowest numbers of vernacular dwellings that are abandoned or in a poor state of conservation.

On the one hand, the situation of vernacular architecture in municipalities that have seen the lowest levels of depopulation is somewhat critical. These municipalities have lost the most architecture, and any architecture built before 1950 remaining is in the poorest state of conservation. If the situation of these case studies is not urgently remedied, these municipalities could lose almost 50% of their vernacular architecture. This entails major cultural problems including loss of local identity and local construction techniques falling into oblivion, as well as the loss of points of reference for the study of local architecture. This situation suggests future lines of research in areas where depopulation has not been so intense, with the aim of identifying the social and political reasons that have brought about the current panorama of vernacular architecture in these territories.

On the other hand, the threat is not so severe for the vernacular architecture of the case studies most affected by depopulation. Figures in relation to the percentage of abandoned dwellings and in a poor state of conservation are low, which, on a more positive note, means that the number of dwellings urgently seeking intervention is low. Thus, small municipalities have a greater capacity for intervention in their vernacular architecture at risk. It should also be remembered that, on average, these municipalities have already lost 40% of their vernacular architecture, so that increasing this loss could lead to the same cultural problems as in municipalities with less depopulation.

However, these results do not mean that the vernacular architecture of the municipalities with greater depopulation is completely safe, as other vulnerability factors also threaten it. This article has shown the analysis in relation to the physical vulnerability factors. However, there are political, demographic, economic and legislative vulnerability factors that must also be analyzed in order to understand the overall risks affecting vernacular architecture. Likewise, in order to corroborate the statistical data, an action plan will be further developed and fieldwork conducted in specific case studies to assess the current

state of vernacular architecture. These are future lines of research to be addressed in the study of threats, risks and resilience factors of vernacular architecture in depopulated areas.

In conclusion, the resilience of vernacular architecture should be recognized. In this regard, it should be considered that 60% of vernacular architecture conserved can promote sustainable development in areas affected by depopulation. This development is possible through considered rehabilitation and restoration actions [77], and the connection of local and seasonal populations in the process of rehabilitating vernacular dwellings [76]. Guaranteeing the correct conservation of vernacular dwellings requires horizontal and vertical cooperation between citizens and administrations.

In this respect, the administrations play a key role in the management of the housing stock, as they can acquire empty dwellings and rehabilitate them as social housing, thus preventing them from being declared ruinous and being demolished. In this way, administrations can influence the conservation of vernacular architecture, as administrators as well as developers. Equally, the recovery of an empty dwelling and its rehabilitation respecting the vernacular surroundings creates employment, increases social housing that is negligible in smaller nuclei [78], and restores integrity to the municipality. In addition, it can consolidate populations in need of a rehabilitated dwelling and attract new populations (as the housing market, which is very small and lacking in transparency in small municipalities, grows [78]). Finally, it also serves as an example of restoration in the hope of encouraging the rest of the inhabitants to carry out respectful interventions on their vernacular dwellings. In short, correct conservation of vernacular architecture using local and traditional materials and techniques can promote the sustainable development of municipalities affected by depopulation, ultimately reducing emigration from the municipality [52].

In order to achieve these objectives, the implementation of a series of actions and measures is needed to curb the negative consequences on vernacular architecture of the abandonment of the rural environment and the growing inequality between the environment and urban areas [79]. Encouraging this type of action and recovery techniques and materials requires the coordination of different public administrations at all levels—municipal, regional and national. At the same time, it is necessary to coordinate the different systems for managing vernacular architecture: protection catalogs, urban planning regulations and licensing processes. To ensure that these systems truly promote the conservation of vernacular architecture, specialist technicians must take part in their creation, also providing clear guidelines for intervention, which will help guide local plans and actions for the control of vernacular heritage [80]. Therefore, the training of specialists in vernacular architecture should be promoted. These specialists should include not only technicians (architects and engineers) but also craftspeople, workers, construction companies and material suppliers. Finally, awareness-raising campaigns geared toward students, communities and local inhabitants could aid in the education and promotion of the elements that define local identity, ultimately aiming to bring about a positive change in the relationship between the population and its vernacular heritage [51]. To summarize, once vernacular architecture is valued and recognized, its rehabilitation, conservation and enhancement can reverse the demographic trends in areas affected by depopulation.

**Author Contributions:** Conceptualization, E.T.M., C.M. and F.V.L.-M.; methodology, E.T.M. and C.M.; formal analysis, E.T.M.; data curation, E.T.M.; writing—original draft preparation, E.T.M.; writing—review and editing, E.T.M., C.M. and F.V.L.-M.; project administration, E.T.M., C.M. and F.V.L.-M.; funding acquisition, E.T.M., C.M. and F.V.L.-M. All authors have read and agreed to the published version of the manuscript.

**Funding:** This research was supported by "Subvenciones para la contratación de personal investigador predoctoral (ACIF/2021/421) financiado por la Generalitat Valenciana y el Fondo Social Europeo" and is part of the research project "RE-HABITAT—Restauración y rehabilitación sostenible de viviendas tradicionales en contextos históricos" funded by the R&D&i programme of the Department for Innovation, Universities, Science and Digital Society of the Generalitat Valenciana (CIAICO/2022/035).

**Data Availability Statement:** The data supporting the conclusions of this article will be made available by the authors on request. These data were derived from the following resources available in the public domain: Instituto Nacional de Estadística (INE): https://www.ine.es/uc/ls0Wp3UCi1 (accessed on 1 March 2024).

**Conflicts of Interest:** The authors declare no conflicts of interest.

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
