# Peer review of "Assessment Methodology for Physical Vulnerability of Vernacular Architecture in Areas Affected by Depopulation: The Case of Comunidad Valenciana, Spain"

_land, doi:10.3390/land13050695_

Round 1
Reviewer 1 Report
Comments and Suggestions for Authors
In my opinion, the introduction would be even richer if the context were expanded to include other areas outside Spain that have endured similar processes of depopulation. Italy would be a natural example, but a glimpse to other, more diverse contexts would be also interesting.
In order to see more clearly the political divisions as well as enlarge the legend, I'd rotate horizontally the Comunidad Valenciana maps. You could also insert a northing symbol to let the reader know of such a rotation of the maps.
Section 2.3 should be restructured and, if possible, partially tabulated. Particularly, point #3 "The state of conservation of the dwelling", would be better presented if summarized and/or tabulated with each of the categories and their descriptions.
References should be shortened by using hyphens (-).
The discussion invites to some reflections. Again, as with the introduction, the discussion should frame the results within a larger context, establishing contrasts as well as possible solutions. There is not a single mention about possible participation from the private sector in helping with channeling investments into those depopulated places. I really do not know what would be the public reaction if administrations decide to invest public funds away from where people currently lives. Universities, forensic engineering programs, and architecture schools could also be part of the solution to keep alive this historical stock of structures.
Also, the municipalities that have not experienced a sharp population decline but, nevertheless, is where vernacular architecture seems to be more in danger, call, in my opinion, for the exploration of a whole new research line. What does the people inhabiting those places think about their dwellings? Do they wish to keep them as they are? Are they concerned by the historical character of the built environment that surrounds them? These questions are not obviously part of your research, but just a provocation stemming from the discussion section of your manuscript.
The wording of the methodology as well as the results are in general terms very well, but the analysis and the contextualization could be expanded and deepened. As I stated before, the presentation of the methodology, could be enhanced.
Comments on the Quality of English LanguageWatch for long sentences. I would advice to be as concise as possible. However, it is well written. There are very few errors. I'd recommend to read it out loud a couple of times to ensure quality.
Author Response
May 8th, 2024
Dear Reviewer,
We send the revisions made for the article entitled “ASSESSMENT METHODOLOGY FOR PHYSICAL VULNERABILITY OF VERNACULAR ARCHITECTURE IN AREAS AFFECTED BY DEPOPULATION: THE CASE OF COMUNIDAD VALENCIANA, SPAIN” based on the observations provided by the reviewer 1. We thank the reviewer for their contribution, which have undoubtedly allowed us to improve our article.
REVIEW REPORT 1:
-In my opinion, the introduction would be even richer if the context were expanded to include other areas outside Spain that have endured similar processes of depopulation. Italy would be a natural example, but a glimpse to other, more diverse contexts would be also interesting.
Thank you for your suggestion. While the analysis of individual country statistics may also result in a parallel investigation, this suggestion has been considered useful and the demographic trends in rural areas of the world, Europe, and certain European countries have been added.
-In order to see more clearly the political divisions as well as enlarge the legend, I'd rotate horizontally the Comunidad Valenciana maps. You could also insert a northing symbol to let the reader know of such a rotation of the maps.
We appreciate your input, however, interpreting a large-scale plan is always easier if it follows the usual direction (facing north). Therefore, for ease of reading, it is considered preferable to leave the position of the plan as it is. Nevertheless, the suggestion to enlarge the legend has been considered useful and has been carried out.
-Section 2.3 should be restructured and, if possible, partially tabulated. Particularly, point #3 "The state of conservation of the dwelling", would be better presented if summarized and/or tabulated with each of the categories and their descriptions.
Thank you for your suggestion, Section 2.3 has been restructured, and each point has been converted into a sub-section. At the same time the paragraphs have been tabulated to improve the understanding of each vulnerability factor.
-References should be shortened by using hyphens (-): DONE
-The discussion invites to some reflections. Again, as with the introduction, the discussion should frame the results within a larger context, establishing contrasts as well as possible solutions. There is not a single mention about possible participation from the private sector in helping with channeling investments into those depopulated places. I really do not know what the public reaction would be if administrations decide to invest public funds away from where people currently lives. Universities, forensic engineering programs, and architecture schools could also be part of the solution to keep alive this historical stock of structures.
This comment has been taken into account and a final paragraph has been added to summarise the actions considered for the reversal of the trend of vernacular architecture conservation and depopulation. As already detailed in the discussion, the research continues with the analysis of other vulnerability factors that show the need for the actions mentioned in the added paragraph.
-Also, the municipalities that have not experienced a sharp population decline but, nevertheless, is where vernacular architecture seems to be more in danger, call, in my opinion, for the exploration of a whole new research line. What does the people inhabiting those places think about their dwellings? Do they wish to keep them as they are? Are they concerned by the historical character of the built environment that surrounds them? These questions are not obviously part of your research, but just a provocation stemming from the discussion section of your manuscript.
This suggestion has been considered and a sentence has been included in the discussion regarding future lines of research for these case studies. However, the discussion already mentioned that the research continues with the analysis of other vulnerability factors such as political, economic, demographic and legislative. This future analysis will include the social and demographic aspect that may affect vernacular architecture.
-The wording of the methodology as well as the results are in general terms very well, but the analysis and the contextualization could be expanded and deepened. As I stated before, the presentation of the methodology, could be enhanced.
Your comment have been taken into account, and both the introduction and discussion have been expanded according to the recommendations made in the review.
Thank you for your review.
Reviewer 2 Report
Comments and Suggestions for Authors
The phenomenon of depopulation in rural areas is very relevant and has not been as much addressed as natural risks. Linked with the vernacular architecture, it is an important topic that must be discussed. The study cases, especially in a country like Spain with very serious problems of depopulation, are very interesting and suitable for this article and all the data obtained looks very accurate.
However, the methodology proposed (In fact, I would say that this is a method and not a methodology) presents many inconsistencies and a better case should be made. It’s very important to justify the selection of these three indicators and why they were chosen. I couldn’t find references in the text related with the method presented or previous research done by the authors. For instance, the second and the third indicators can sometimes be redundant, since and abandoned (empty) building probably will have a deficient or poor state of conservation.
Regarding the state of conservation, how was the evaluation performed and which were the guidelines? In the section 2.3. some indications are given but it is not clear if there was a fieldwork campaign since section 2.4. only references the treatment of quantitative and statistical data.
I would recommend to check some articles that have proposed or utilized vulnerability index methods for the analysis of seismic vulnerability of vernacular architecture.
https://doi.org/10.1016/j.engstruct.2019.109381
https://doi.org/10.1080/13632469.2019.1657987
https://doi.org/10.1080/15583058.2022.2070049
While these other methods are normally designed for earthquake vulnerability, they consider many other parameters or indicators, and in the end, they explore the physical vulnerability of vernacular buildings. In the method proposed in this article some important aspects are ignored, mainly the construction materials and systems or others like geometrical and morphological relations of the buildings. The physical vulnerability should entail many aspects and this paper is focusing on some statistical data and the only risk analyzed is the abandonment of buildings by depopulation.
I also would recommend to increase and improve the bibliography, since most of the literature is in Spanish and is not very up to date. In this case it is understandable to have some local references due to the depopulation and abandonment issues exposed but a scientific paper must complete a rigorous state of the art with high impact global literature. It would be interesting to add more global (and also recent) references to the phenomenon of depopulation and especially physical vulnerability.
I would also recommend to revise the citation format and while they are not mandatory, it would be interesting to add some conclusions and discuss the following actions and research interests.
Comments on the Quality of English LanguageMinor editing of English language required.
Author Response
May 8th, 2024
Dear Reviewer,
We send the revisions made for the article entitled “ASSESSMENT METHODOLOGY FOR PHYSICAL VULNERABILITY OF VERNACULAR ARCHITECTURE IN AREAS AFFECTED BY DEPOPULATION: THE CASE OF COMUNIDAD VALENCIANA, SPAIN” based on the observations provided by the reviewer 1. We thank the reviewer for their contribution, which have undoubtedly allowed us to improve our article.
REVIEW REPORT 2:
-The phenomenon of depopulation in rural areas is very relevant and has not been as much addressed as natural risks. Linked with the vernacular architecture, it is an important topic that must be discussed. The study cases, especially in a country like Spain with very serious problems of depopulation, are very interesting and suitable for this article and all the data obtained looks very accurate.
However, the methodology proposed (In fact, I would say that this is a method and not a methodology) presents many inconsistencies and a better case should be made. It’s very important to justify the selection of these three indicators and why they were chosen. I couldn’t find references in the text related with the method presented or previous research done by the authors. For instance, the second and the third indicators can sometimes be redundant, since and abandoned (empty) building probably will have a deficient or poor state of conservation.
The authors would like to thank the reviewer for the kind comments regarding the interest and suitability of this research, as well as the accuracy of the data. This research has been carried out as part of a research project. The authors have a certain experience of research into the values and risks of vernacular architecture, among which they have identified depopulation as a risk (see the references mentioned below). For this reason, this research has been proposed, aiming to present an in-depth analysis of all the vulnerability factors associated with the risk of depopulation.
- Mileto, C.; Vegas López-Manzanares, F. Conservation and Restoration of Traditional Architecture. In VERSUS+: Heritage for people. Sharing vernacular knowledge to build the future; Dipasquale, L., Mecca, S., Montoni, L., Eds.; Firenze University Press: Firenze, 2023; pp. 81–85.
- Mileto, C.; Vegas, F.; Cristini, V.; García-Soriano, L. Traditional Spanish Architecture “on the Edge”: An Analysis of Benchmarks Related to Conservation Policies. In Villages et quartiers à risque d’abandon; Hadda, L., Mecca, S., Pancani, G., Carta, M., Fratini, F., Galassi, S., Pittaluga, D., Eds.; Firenze University Press: Firenze, 2022; Vol. 3, pp. 83–90.
- Cristini, V.; Baró Zarzo, JL.; Mileto, C.; Vegas López-Manzanares, F.; Caruso, M.; Tortajada Montalva, E. (2022). “For sale: empty Spain” Raising awareness on abandoned buildings and depopulated villages. En Proceedings HERITAGE 2022 - International Conference on Vernacular Heritage: Culture, People and Sustainability. Editorial Universitat Politècnica de València. 553-558. https://doi.org/10.4995/HERITAGE2022.2022.15768
To justify the selection of the three indicators, section 2.3. Physical vulnerability indicators have been further developed.
The research does not consider information on abandoned buildings and buildings in a poor state of conservation to be redundant. This statement is supported by the fieldwork and bibliography, which have shown that on many occasions abandoned buildings are the ones that best preserve the vernacular architecture, given that buildings in use, both year-round and seasonal, have modified the original elements through interventions with industrial materials, extensions and demolition of significant parts such as the roofs. Therefore, abandoned buildings can be considered to be at risk, as a lack of maintenance can trigger a significant degradation process. Furthermore, these data provide results on the number of buildings in this situation that can be acquired for rehabilitation. However, this does not necessarily imply that it is currently in a poor state of conservation, but rather that it is at future risk. Finally, the results show that these are not redundant vulnerability factors, since the percentages of each factor are not the same (21% of the dwellings are abandoned and 28% are in a poor state of conservation), while the level of risk of each factor on the case studies are also different (as seen in figures 8 and 11).
-Regarding the state of conservation, how was the evaluation performed and which were the guidelines? In the section 2.3. some indications are given but it is not clear if there was a fieldwork campaign since section 2.4. only references the treatment of quantitative and statistical data.
This point is addressed in the first two paragraphs of section 2.4 Data and methodology, where it was stated that the statistical information for the type of use, year of construction and state of conservation of the dwelling was obtained from the 2011 census. However, in order to make the methodology adopted more comprehensive, further information has been provided on the collection of information.
-I would recommend to check some articles that have proposed or utilized vulnerability index methods for the analysis of seismic vulnerability of vernacular architecture.
https://doi.org/10.1016/j.engstruct.2019.109381
https://doi.org/10.1080/13632469.2019.1657987
https://doi.org/10.1080/15583058.2022.2070049
While these other methods are normally designed for earthquake vulnerability, they consider many other parameters or indicators, and in the end, they explore the physical vulnerability of vernacular buildings. In the method proposed in this article some important aspects are ignored, mainly the construction materials and systems or others like geometrical and morphological relations of the buildings. The physical vulnerability should entail many aspects and this paper is focusing on some statistical data and the only risk analysed is the abandonment of buildings by depopulation.
Thank you for the recommendations. These have been revised in order to be able to apply part of the methodology. However, the scale of the analysis of the articles does not correspond to the scale proposed in this research. The investigation focuses on statistical data on vulnerability factors analysed in a generalised way for 180 case studies and more than 30,000 vernacular dwellings. This quantitative analysis aims to create a rapid analysis methodology for a macro-territorial scale, providing results that are easily comparable between case studies, and allowing the creation of an order of urgency so that administrations can act according to the level of risk. This research does not only take into account the abandonment of the buildings, but also the state of conservation assigned by the Spanish public administration. The loss of vernacular architecture in each case study has also been quantified, as well as its representativeness in the total number of dwellings in the municipalities. All these statistical vulnerability factors not only allow an order of action to be established but would also enable the figures of the current state of vernacular architecture in each case study to be understood. This allows a preliminary analysis strategy to be set up before fieldwork, so that efforts can then be focused on understanding further physical factors affecting individual case studies. In order to avoid confusion, and thanks to the suggestion made, a paragraph has been added at the end of Chapter 2.4. Lastly, it was already included in the discussion that the research continues with the analysis of other vulnerability factors such as political, economic, demographic and legislative. This future analysis will include other aspects that may affect vernacular architecture.
-I also would recommend to increase and improve the bibliography, since most of the literature is in Spanish and is not very up to date. In this case it is understandable to have some local references due to the depopulation and abandonment issues exposed but a scientific paper must complete a rigorous state of the art with high impact global literature. It would be interesting to add more global (and also recent) references to the phenomenon of depopulation and especially physical vulnerability.
Taking into account the suggestion made, several updated references combining vernacular architecture and depopulation have been added:
Mecca, S. La Régénération Des Villages Est Un Élément d’un Projet de Croissance Durable et Équitable. In Villages et quartiers à risque d’abandon; Hadda, L., Mecca, S., Pancani, G., Carta, M., Fratini, F., Galassi, S., Pittaluga, D., Eds.; Firenze University Press: Firenze, 2022; Vol. 1, pp. 28–33
Haridi, F.-Z.; Boulemaredj, A.; Laouier, A.E.; Ouled-Diaf, A.; Saïfi, A. Abandon de Ksour Sahariens Entre Indifférence et Désintéressement. In Villages et quartiers à risque d’abandon; Hadda, L., Mecca, S., Pancani, G., Carta, M., Fratini, F., Galassi, S., Pittaluga, D., Eds.; Firenze University Press: Firenze, 2022; Vol. 3, pp. 228–237.
Domínguez Álvarez, J.L. La Despoblación En Castilla y León: Políticas Públicas Innovadoras Que Garanticen El Futuro de La Juventud En El Medio Rural. Cuad. Investig. en Juv. 2019, 6, doi:10.22400/cij.6.e028.
de las Rivas Sanz, J.L.; Fernández Maroto, M.; Martínez Sierra, M.; Rodrigo González, E. Análisis Sobre Los Principios de Ordenación y Los Instrumentos Urbanísticos Para La Protección de La Arquitectura Tradicional En Pequeños Municipios Rurales de España; Subdirección General del Instituto del Patrimonio Cultura de España, Ed.; Ministerio de Educación, Cultura y Deporte, 2016;
Magnaghi, A. Forma de Metrópoli y Desterritorialización. In El proyecto local. Hacia una conciencia del lugar; Universitat Politècnica de Catalunya: Barcelona, 2011; pp. 69–82 ISBN 978-84-7653-928-6.
Lalana Soto, J.L.; Pérez-Eguíluz, V. In a Village of Castilla... Dealing with Heritage Conservation in a Depopulation Context. 2° Convegno Internazionale sulla Doc. Conserv. e Recuper. del Patrim. Archit. e sulla tutela paesaggistica. 2014, 1431–1436.
-I would also recommend to revise the citation format and while they are not mandatory, it would be interesting to add some conclusions and discuss the following actions and research interests.
Thank you for your feedback. The citation format has been revised and shortened by using hyphens.
Regarding the conclusions, new aspects have been added that must be taken into account to ensure the proper conservation of vernacular architecture, and following actions to corroborate the statistical data.
Thank you for your review.
Reviewer 3 Report
Comments and Suggestions for Authors
Dear Authors,
congratulations on your paper. Here are some minor revisions.
Line 97. Revise punctuation (a dot after territory)
Line 103. It is advisable to add highlighted text as follows In terms of depopulation and rural dwellings, many factors, including the insalubrious conditions of rural dwellings lead to an increased risk of emigration from the countryside to the city
Par 2.3. Clarify the relationship between kinds of use (or lack of use) and physical conditions. Consider also qualitative aspects of use (in fact, use in itself is no guarantee of conservation: a different matter is the case of sustainable use of the architectural heritage).
Par. 3.1 Clarify whether all buildings constructed before 1950 in the case study area belong to the category of vernacular constructions, or whether they are the majority
Regarding References.
1) Avoid citing edited works, but rather specific texts in edited volumes (for instance [50])
2) Rudofsky's work is cited twice, once in the original version and once in the Spanish version. Check whether it is appropriate to differentiate the citation
line 97: revise punctation
line 103: the causes of the risks of emigration from the countryside to the city are a combination of factors (and not only the insalubrious conditions of vernacular dwellings). The Authors could provide a concise picture of the causes of depopulation, both general and in the specific study area
Author Response
May 8th, 2024
Dear Reviewer,
We send the revisions made for the article entitled “ASSESSMENT METHODOLOGY FOR PHYSICAL VULNERABILITY OF VERNACULAR ARCHITECTURE IN AREAS AFFECTED BY DEPOPULATION: THE CASE OF COMUNIDAD VALENCIANA, SPAIN” based on the observations provided by the reviewer 1. We thank the reviewer for their contribution, which have undoubtedly allowed us to improve our article.
REVIEW REPORT 3:
-Line 97. Revise punctuation (a dot after territory): DONE
-Line 103. It is advisable to add highlighted text as follows In terms of depopulation and rural dwellings, many factors, including the insalubrious conditions of rural dwellings lead to an increased risk of emigration from the countryside to the city: DONE
-Par 2.3. Clarify the relationship between kinds of use (or lack of use) and physical conditions. Consider also qualitative aspects of use (in fact, use in itself is no guarantee of conservation: a different matter is the case of sustainable use of the architectural heritage).
Thank you for your suggestion. The relationship between the kinds of use and the risks to vernacular housing has been added to Par 2.3.2. This takes the form of a summarised presentation, as the socio-cultural aspects are analysed in another section of the research, but this is not the aim of this article. The results on the specific issue of seasonal population will be developed together with the demographic risk of vernacular architecture which will be presented later in further articles, as explained in the conclusions.
- Par. 3.1 Clarify whether all buildings constructed before 1950 in the case study area belong to the category of vernacular constructions, or whether they are the majority.
Your comment has been considered and the rationale for the choice to establish all dwellings built before 1950 as vernacular architecture has been justified in Par 2.3.1.
-Regarding References.
1) Avoid citing edited works, but rather specific texts in edited volumes (for instance [50]). DONE
Citations to the specific chapter of several publications have been modified:
Lewcock, R. Westernization and Cultural Interaction. In Encyclopedia of Vernacular Architecture of the World; Oliver, P., Ed.; Cambridge University Press: Cambridge, 1997; pp. 121–122.
Oliver, P. Introduction. In Encyclopedia of Vernacular Architecture of the World; Oliver, P., Ed.; Cambridge University Press: Cambridge, 1997; pp. xxi–xxviii.
Mileto, C.; Vegas, F. Heritage for People. A Project for Connecting People with Their Tangible and Intangible Heritage. In VERSUS+: Heritage for people. Sharing vernacular knowledge to build the future; Dipasquale, L., Mecca, S., Montoni, L., Eds.; Firenze University Press: Firenze, 2023; pp. 17–29 ISBN 978-88-3338-200-5.
2) Rudofsky's work is cited twice, once in the original version and once in the Spanish version. Check whether it is appropriate to differentiate the citation. DONE.
Thank you for the suggestion, the error has been corrected. The quotation has been changed back to the original format.
-line 97: revise punctation: DONE
-line 103: the causes of the risks of emigration from the countryside to the city are a combination of factors (and not only the insalubrious conditions of vernacular dwellings). The Authors could provide a concise picture of the causes of depopulation, both general and in the specific study area: DONE
Thank you for your review.
Round 2
Reviewer 2 Report
Comments and Suggestions for Authors
Thanks to the authors for addressing all the comments and suggestions . This article could be a used as an interesting frame of reference for the vulnerability assessment of vernacular architecture affected by depopulation processes.